# OpenReview forum: "ACAV-1M: Data Curation and Benchmarking for Audio-Visual Representation Learning"
_ICLR.cc/2025/Conference — Submitted to ICLR 2025_

### Official Review · Reviewer_vKz8 · 2024-10-22

**Soundness:** 2
**Presentation:** 2
**Contribution:** 2
**Rating:** 3
**Confidence:** 4

**Summary:**

The paper introduces a new large-scale audio-visual dataset, ACAV-1M, designed to enhance multimodal learning models by accurately aligning audio and video information. It provides a detailed description of the dataset construction process, covering the generation of text descriptions, data filtering, and alignment, followed by benchmarking on several downstream tasks.

**Strengths:**

1. The authors implement a comprehensive alignment process, leveraging a multimodal large language model and ImageBind to ensure semantic, instance-level, and temporal consistency between audio, video, and text. This significantly improves the quality of the dataset.
2. The models trained on ACAV-1M demonstrate superior performance in tasks such as audio-visual classification, source separation, and question-answering. These models outperform existing approaches across multiple evaluation metrics, highlighting the potential of ACAV-1M in advancing multimodal learning.

**Weaknesses:**

1. There are inconsistencies in the writing that hinder the paper’s readability. For example, in the caption for Figure 1, the authors refer to a "Large Language Model (LLM) (OpenAI, 2023) to generate multiple captions." However, in Section 3.1 on data curation, they mention "VideoLLaVA (Lin et al., 2023) generates several sentence-level descriptions." These contradictions create confusion and make the paper harder to follow. Furthermore, the structure of Section 3.2 could be clarified to improve the overall flow.
2. The authors use the open-source Video-LLaVA framework to generate comprehensive captions for audio-visual content, which forms a critical part of their work. However, the Video-LLaVA framework does not include an audio encoder and processes only images and video. The paper does not clearly explain where or how the audio captions are generated, which raises concerns about whether it is appropriate to use Video-LLaVA for audio-visual captioning. This is a key technical issue that should be addressed.
3. The paper mentions that each audio segment is standardized to a 10-second duration. However, it does not provide sufficient detail on how these segments are cut or how the semantic integrity of the data is maintained after cutting. It is unclear whether the alignment between audio and video is preserved during this process. A more thorough analysis of these technical aspects would strengthen the paper and reassure readers about the dataset's quality.
4. The paper employs automated alignment and filtering techniques to evaluate the synchronization between audio, video, and text. While this approach is efficient, it may not be equally effective for all types of content. For example, music or ambient sounds may not require perfect alignment with visual elements. The authors could consider adjusting filtering parameters based on the content type to improve the alignment process for different scenarios.
5. While the alignment process enhances data quality, overly strict filtering thresholds may cause valuable samples to be discarded. The paper does not discuss or quantify potential data loss resulting from this filtering, which could lead to a dataset biased towards content that is "easy to align." This reduction in sample diversity might limit the dataset's utility in certain tasks. The authors should explore a balance between strictness and diversity, or at least provide an analysis of the data loss caused by filtering.

**Questions:**

1. While ACAV-1M performs well in experimental tasks, is this improvement primarily due to better data alignment from the high threshold used in filtering, or does it stem from increased data diversity and quality?
2. Although the model excels across many tasks, are there specific scenarios or tasks, such as long video processing or audio-video alignment in noisy environments, where performance bottlenecks occur?
3. The paper mentions using large language models to generate text descriptions for both audio and video, but large language models are known to exhibit biases. Did the authors take any steps to detect and mitigate these biases during the dataset creation process?

---

### Official Review · Reviewer_GrEK · 2024-10-27

**Soundness:** 3
**Presentation:** 3
**Contribution:** 3
**Rating:** 6
**Confidence:** 3

**Summary:**

The paper presents the ACAV-1M dataset, consisting of one million high alignment audio-visual samples. A novel data curation pipeline transforms raw video and audio into detailed, aligned captions using a multimodal large language model. Comprehensive benchmarks and task-specific methods leverage this dataset to advance audio-visual learning, supported by extensive experiments showing its effectiveness and scalability compared to existing datasets.

**Strengths:**

- The method for data filtering takes into account the critical issue of audio-visual temporal alignment. This temporal alignment could facilitate more fine-grained downstream tasks.

- The paper employs numerous well-designed downstream tasks to demonstrate the audio-visual alignment in the ACAV-1M dataset, and the results are convincing.

**Weaknesses:**

Although the paper demonstrates the strength of the audio-visual alignment through various downstream tasks, could more intuitive representations be provided? For instance, a statistical similarity comparison calculated via ImageBind.

**Questions:**

- The paper includes video captioning. What effect might be observed if AudioLLM were used for audio captioning followed by a similar similarity comparison?
- The training set of ImageBind is Audioset, which, as mentioned, does not have strong audio-visual alignment. As a hypothesis, if ImageBind were retrained using ACAV-1M, and a new representation extractor were used for filtering, could this lead to improved results?

---

### Official Review · Reviewer_hSaS · 2024-11-02

**Soundness:** 2
**Presentation:** 2
**Contribution:** 2
**Rating:** 5
**Confidence:** 5

**Summary:**

The paper presents ACAV-1M, a dataset of one million curated audio-visual samples, designed to improve representation learning in models by ensuring precise audio-visual alignment. Through a rigorous data curation pipeline, ACAV-1M enhances the natural synchronization of audio and visual information, leveraging a multimodal large language model for text captioning and filtering for temporal and instance alignment. This paper argues the ACAV-1M can be used as a pre-trained dataset for various tasks, including classification, source localization, audio-visual retrieval, and so on.

**Strengths:**

1. The paper is well written.
2. The motivation of this paper is reasonable. Large-scale audio-visual datasets are eagerly needed by the community

**Weaknesses:**

1. The data collection paradigm uses ImageBind to measure the alignment between audio-visual-language. However, as discussed in [1], ImageBind actually is inferior to aligning audio and language. Better models or more in-depth comparative discussions are needed.
2. The main baseline in the paper should be AudioSet, which includes twice the amount of data and has been widely used. Table 1 claims that ACAV-1M can support more tasks, while AudioSet cannot. However, in actual experiments, ACAV-1M is mostly used as pre-training data, which can also be replaced by AudioSet, so the claim in Table 1 is not convincing. A fairer comparison with AudiosSet is needed to highlight the necessity of ACAV-1M.
3. Lack of some more advanced baselines, such as using ImageBind for audio-visual retrieval.

[1] FreeBind: Free Lunch in Unified Multimodal Space via Knowledge Fusion[C]//Forty-first International Conference on Machine Learning.

**Questions:**

See Weakness.

---

### Official Review · Reviewer_VqVY · 2024-11-03

**Soundness:** 2
**Presentation:** 3
**Contribution:** 2
**Rating:** 3
**Confidence:** 4

**Summary:**

In many existing audio-visual datasets, there are numerous instances where audio and video signals are misaligned, such as in samples with background music. This paper proposes a large-scale audio-visual dataset and introduces a pipeline to ensure the alignment and temporal synchronization of audio and video signals. The authors also conduct experiments across multiple tasks, establishing baselines.

**Strengths:**

- The paper validates its approach across various tasks, demonstrating promising results.

- The audio-visual dataset that ensures the audio-visual correspondence and synchronization of samples in the dataset will be helpful.

**Weaknesses:**

- Addressing the issue of audio-video misalignment is crucial, as most misaligned samples in common datasets involve voice-over narration and background music that lack synchronization with the visual content. In a random inspection of samples from the video list provided by the authors, I identified a lot of cases with audio-video misalignment, such as “kCb7VJgc1Zc,” “_QQHFllvooU,” and “JUChESSdblA.” These samples include voice narration but lack a visible talking head in the video, which generally suggests inconsistency between the audio and video content. This raises concerns about whether ACAV-1M can reliably ensure effective alignment of audio and video. Please detailed the quality control process - for instance, what percentage of samples were manually checked, and whether there's a process for addressing misalignments identified after curation.
- The paper does not seem to outline steps to verify the audio-visual consistency of the filtered dataset. This omission leads to an open question: are the model’s performance gains due to the dataset’s size or improved audio-visual alignment? Would a similar boost be observed with models trained on ACAV-100M, even without alignment assurance? Can you compare performance of models trained on ACAV-100M without alignment process and ACAV-1M with alignment process?
- The authors currently do not provide metadata for the dataset. I believe that various types of metadata, such as video captions, timestamps, and cluster categories in AVAC-100M, are essential. Similarity information could allow users to customize subsets and filter the samples they need. As captions are one of the supported modalities in this dataset, they should be included, while timestamps would enable users to segment data more effectively.

**Questions:**

- To my understanding, in VGGSound, the audio-visual and text-audio similarity scores in ImageBind generally cluster around 0.3, with only a few samples reaching 0.5. Could you provide additional similarity distribution plots to illustrate the extent of filtering applied to the dataset?
- Since this paper presents only one model version, it likely relies on a specific model size. Which model did you select—ViT-Base or ViT-Large? Are these settings consistent with prior works, and could you share additional experimental details to clarify?
- Given that ImageBind primarily undergoes training on audio-visual and text-visual pairs, with audio-language similarity emerging as an untrained capability without direct audio-text training, could you discuss the reliability of this filtering approach within these training constraints?

**Details Of Ethics Concerns:**

The authors appear to plan on storing video data on cloud storage to provide a mirror, but it’s uncertain whether this might violate YouTube’s privacy policy: *"The dataset is hosted on a dedicated server managed by our institution, ensuring reliable access and
download speeds. We also provide mirror links through major cloud storage providers to ensure
redundancy and availability."*

---

### Official Review · Reviewer_P5af · 2024-11-11

**Soundness:** 2
**Presentation:** 2
**Contribution:** 1
**Rating:** 3
**Confidence:** 5

**Summary:**

This paper introduces ACAV-1M, a large-scale audio-visual dataset containing one million samples with carefully curated data processing pipeline and comprehensive benchmarks for various audio-visual tasks. The dataset demonstrates superior performance across multiple tasks.

**Strengths:**

The dataset's comprehensive size of one million samples may provide a robust foundation for training audio-visual models.

**Weaknesses:**

Data Distribution Issues:

- Lack of distribution statistics. The paper fails to provide detailed breakdowns of video types, categories, and their proportions in the dataset, making it impossible to assess dataset balance and potential biases.

- Missing temporal distribution. No information about the temporal spread of collected data, which could affect the dataset's representativeness across different time periods.

Data Source Issues:

- Undisclosed sources. The paper doesn't specify the origins of the video data, raising concerns about data quality and reliability. No explanation of how videos were selected or filtered during the initial collection phase.
- Licensing ambiguity. Lack of discussion about copyright and usage rights, potentially limiting the dataset's practical applicability.

Filtering Methodology Concerns:

- Limited performance comparison. The implementation of instance and temporal filtering isn't well-justified, especially given that most professional audio-visual content is naturally synchronized. The study only explores similarity thresholds for filtering. However, there is no comparative analysis between filtered and unfiltered datasets. On the other hand, how much data is filtered out is not reported, and it is unclear whether the filtering process actually unnecessarily reduces the dataset size, therefore, it is hard to judge whether using the unfiltered dataset, which could have much larger size, might yield better results.

- Uniform threshold application. The study applies the same similarity threshold (50%) across language alignment, instance alignment, and temporal alignment. This one-size-fits-all approach may be problematic as different types of alignment may require different thresholds for optimal filtering. The lack of independent threshold optimization for each alignment type could lead to suboptimal filtering results.

Foundation Model Evaluation Concerns:

- MLLM and ImageBind reliability. The paper relies heavily on VideoLLaVA and ImageBind for data processing but lacks rigorous evaluation of these models' performance  (e.g., using human annotated subsets for evaluation):

  - No human evaluation of MLLM-generated captions to verify their accuracy and relevance

  - Missing analysis of ImageBind's cross-modal alignment accuracy on this dataset

**Questions:**

See weaknesses

**Details Of Ethics Concerns:**

Copyright issues may prevent the releasing of dataset.

---

### Meta-Review · Area_Chair_5Cnz · 2024-12-18

**Metareview:**

This paper introduces a large-scale audio-visual dataset (ACAV-1M) aimed at improving multimodal alignment. It employs an intricate curation pipeline and filtering for better semantic integrity. While the authors highlight advances over existing datasets and show improved performance on various tasks. key weaknesses remain. The submission’s claims largely rely on experiments without sufficient independent validation. Some concerns from initial reviews, such as practical impact and external comparability, remain insufficiently addressed. Overall, despite the efforts to clarify methods and mitigate biases, the claimed benefits remain unconvincing. Therefore, I recommend rejection.

**Additional Comments On Reviewer Discussion:**

In response, the authors provided more details on their alignment methodology, clarified the captioning workflow, and added comparative experiments (e.g., ACAV-1M vs. ACAV-100M, alternative alignment models). They also addressed bias and ethical issues. Although these revisions offered some improvement, the lack of broader reviewer engagement left important concerns unresolved. The updates, did not sufficiently alter the final decision to reject.

---

### Decision · Program_Chairs · 2025-01-22

Reject